# Jugular Foramen Tumors: Surgical Strategies and Representative Cases

**DOI:** 10.3390/brainsci14020182

**Published:** 2024-02-17

**Authors:** Andrea L. Castillo, Ali Tayebi Meybodi, James K. Liu

**Affiliations:** 1Department of Neurological Surgery, New Jersey Medical School, Newark, NJ 07103, USA; at1085@njms.rutgers.edu; 2Department of Neurosurgery, Cooperman Barnabas Medical Center, RWJ Barnabas Health, Livingston, NJ 07039, USA; 3Skull Base Institute of New Jersey, Neurosurgeons of New Jersey, Livingston, NJ 07039, USA

**Keywords:** jugular, tumor, foramen, paraglanglioma, combined

## Abstract

(1) Background: Jugular foramen tumors are complex lesions due to their relationship with critical neurovascular structures within the skull base. It is necessary to have a deep knowledge of the anatomy of the jugular foramen and its surroundings to understand each type of tumor growth pattern and how it is related to the surrounding neurovascular structures. This scope aims to provide a guide with the primary surgical approaches to the jugular foramen and familiarize the neurosurgeons with the anatomy of the region. (2) Methods and (3) Results: A comprehensive description of the surgical approaches to jugular foramen tumors is summarized and representative cases for each tumor type is showcased. (4) Conclusions: Each case should be carefully assessed to find the most suitable approach for the patient, allowing the surgeon to remove the tumor with minimal neurovascular damage. The combined transmastoid retro- and infralabyrinthine transjugular transcondylar transtubercular high cervical approach can be performed in a stepwise fashion for the resection of complex jugular foramen tumors.

## 1. Introduction

Tumors found in the jugular foramen are complex to approach due to the intricate surrounding neurovascular anatomy of the craniocervical junction [1]. Paragangliomas are the most frequently found tumors in the jugular foramen, with schwannomas following closely behind. Other less common tumors found in this area include meningiomas, chordomas, chondrosarcomas, and plasmacytomas. Metastases and malignant tumors that originate in nearby anatomical structures like the nasopharynx, parotid, and temporal bone can also spread to the jugular foramen during later stages [1]. Lastly, endolymphatic sac tumors can potentially extend to the jugular foramen and originate from the posterior medial region of the petrous bone [1,2].

The jugular foramen is a hiatus between the temporal and occipital bones [3]. The petrous portion of the temporal bone forms its anterolateral margin, and the occipital bone’s condylar part forms its posteromedial margin [4]. The jugular foramen is at the crossroads of the caudal cranial nerves and the sigmoid and inferior petrosal sinuses with the otic capsule and lower brainstem in close proximity. The internal carotid artery is related to it anteromedially. These anatomic relationships pose a genuine challenge from a surgical perspective [3,5,6,7].

It is crucial to have a deep knowledge of the anatomy of the jugular foramen and its surroundings to comprehend each tumor growth pattern and its relation to the neurovascular structures surrounding them. These essential tools will allow the neurosurgeon to develop the appropriate surgical techniques to remove these tumors safely and effectively.

This work aims to provide a guide with the primary surgical approaches to the jugular foramen and familiarize neurosurgeons with the anatomy of the region. Although there are a variety of surgical approaches, it is paramount to individualize each case and tailor the approach for the patient to provide the appropriate necessary exposure to remove the most amount of tumor with the least amount of neurovascular morbidity.

## 2. Materials and Methods

A comprehensive description of the surgical approaches for jugular foramen tumors is summarized and detailed. Representative cases for different types of tumors were selected based on the senior author’s experience to describe the combined transmastoid retro- and infralabyrinthine high cervical approach used in large jugular foramen tumors. Operative pearls, proper selection approach, and avoidance of complications are also discussed.

## 3. Results

The jugular foramen can be approached through several pathways depending on the tumor’s configuration. Each case must be carefully assessed before selecting the adequate approach. Frequently, subtle nuances distinguish each approach from others, adding complexity for proper understanding. To simplify this, we can divide the approaches based on anterolateral and posterolateral perspectives [8]. The anterolateral approaches include the dissection of the structures located in front and lateral to the sigmoid sinus and jugular foramen, and the posterolateral approaches are accessed via the dissection of structures behind and lateral to the sigmoid sinus and jugular foramen [8]. Every approach has its advantages and disadvantages, and their specific indications for each type of tumor are shown in Table 1. Also, a combined approach can be used, maximizing the advantages from both anterolateral and posterolateral corridors [9].

### 3.1. Posterolateral Approaches

#### 3.1.1. Retrosigmoid Approach

This approach is the workhorse for posterior fossa lesions. It is the most common and well-known approach for jugular foramen tumors in neurosurgery. It is indicated predominantly for intradural tumors with little or no extension to the extradural compartment [10]. A C-shaped retroauricular skin incision is created, posterior and parallel to the outline of the pinna, followed by a lateral suboccipital craniotomy exposing the dura inferior and posterior to the transverse and sigmoid sinuses [3]. The dura is also opened in a C or U shape manner. The cerebellum is gently displaced medially away from the posterior petrous surface of the temporal bone to expose the lateral aspect of the brainstem and intracranial segments of the cranial nerves exiting through the internal acoustic meatus and jugular foramen [3,8] Figure 1A.

Even though it is a straightforward approach, this is unsuitable for large tumors that go out the jugular foramen’s extracranial compartment [11]. A suprajugular extension of the retrosigmoid approach was described by Matsushima et al. in 2014. The area inferior to the internal acoustic meatus, medial to the endolymphatic depression, and the surrounding superior half of the glossopharyngeal dural fold is drilled to access the suprajugular aspect of the jugular foramen. This approach is suitable for tumors extending into the jugular foramen’s upper part and above the jugular bulb [12].

#### 3.1.2. Far-Lateral Approach 

The far-lateral approach is a more inferior extension of the retrosigmoid approach and involves extending further laterally from the lateral suboccipital approach [13,14]. It provides a better access to the foramen magnum and craniocervical junction, extending the corridor inferior and lateral to the lower cranial nerves [15]. Its primary indication is tumors that extend to the foramen magnum anteriorly or laterally to the lower brainstem at the craniocervical junction [13,14]. The jugular foramen is exposed posteriorly [3]. The skin can be incised using a retro auricular C-shaped incision, or by a classic hockey stick incision beginning from the mastoid tip; the line runs posteriorly to the inion and finally ends inferiorly to the spinous process of C2. The muscles attached to the occipital bone are detached en bloc and reflected inferiorly. A retrosigmoid suboccipital craniotomy that unroofs the foramen magnum and C1 hemilaminectomy is performed. It has three variations depending on which area is desired for access: the jugular foramen, the lower clivus, and the premedullary area [8,15,16]. Some lesions located along the anterolateral margin of the foramen magnum might only need a basic far-lateral approach without drilling the condyle (retrocondylar). Nevertheless, the far-lateral approach can provide a route through which the transcondylar, supracondylar, and paracondylar extensions can be completed with a further increase in the working space along the anterior border of the foramen magnum, jugular tubercle area, and posterior margin of the jugular foramen, respectively [15,16,17,18].

The degree of drilling the occipital condyle will depend on the access area needed.

The transcondylar approach is performed by drilling the posterior condyle, which gradually expands the working space to the anterior brainstem and petroclival area [19,20]. The percentage of condyle to be removed can vary from only the posterior one third with no instability of the craniovertebral junction to as far as the posterior half, which will require fixation of the craniovertebral junction due to instability of the atlanto-occipital joint [21,22]. The complete trans-condylar approach during which the posterior two thirds of the condyle is removed will have the hypoglossal canal as the anterior limit and can provide a dramatic increase in petroclival exposure, especially if the jugular tuberculum is also removed [13,20]. This approach is generally used to reach the premedullary area [13,23].

The supracondylar extension is the approach used to reach the jugular tubercle. It involves drilling the occipital bone above and behind the occipital condyle by completely preserving the occipital condyle and including the drilling of the condylar fossa. The condylar fossa contains the posterior condylar canal below, and drilling it results in a defect in the posterior part of the jugular tubercle [8,13,16].

Lesions affecting the posterior section of the jugular foramen may be approached using the paracondylar extension. These can include paragangliomas of the jugular foramen and dumbbell schwannomas of the lower cranial nerves. The approach is tailored towards the jugular process of the occipital bone, which is located lateral to the occipital condyle and is the site of attachment of the rectus capitis lateralis muscle. It involves the skeletonization and opening of the hypoglossal canal, partial drilling of the lateral portion of the occipital condyle, and the mastoid tip [8,13,16,17].

The preservation of the occipital condyle and the C-1 lateral mass, as well as the attachments of the alar and transverse ligaments to the anterior one third of the occipital condyle and the anterior one third of the C-1 lateral mass, are among the key variables that are responsible for maintaining the stability of the occipitocervical junction [15,17]. An occipital-cervical fusion operation is required when the integrity of these structures has been impaired, whether through a complete trans-condylar approach, a transplacental approach, or tumor destruction of these areas. Fusion is usually performed as a second-stage operation and when there is no sign of cerebrospinal fluid leakage [15,17,20].

### 3.2. Anterolateral Approaches

#### 3.2.1. Postauricular Transtemporal Approach

This approach can provide excellent exposure to the jugular foramen and lateral skull base. It is accessed from a lateral approach through the mastoid and the neck (mastoid-neck approach) [8]. A post-auricular C-shaped skin incision provides exposure for a mastoidectomy and neck dissection. The external auditory canal can be either preserved or transected with blind sac closure, depending on the anterior extension of the tumor [3]. The mastoidectomy primarily involves the infralabyrinthine region with exposure of the sigmoid sinus, jugular bulb, and mastoid segment of the facial nerve. Hearing does not have to be sacrificed, and it can be preserved by maintaining the footplate of the stapes. Nevertheless, to fully expose the lateral half of the jugular foramen, the mastoid portion of the facial nerve is mobilized anteriorly (which can cause facial nerve palsy), the styloid process must be resected, and detachment of the rectus capitis lateralis muscle from the jugular process of the occipital bone is performed [8,24,25]. The middle and posterior cranial fossa dura in front (Trautman’s triangle) and behind the sigmoid sinus are exposed [26].

#### 3.2.2. Preauricular Subtemporal Infratemporal Approach

This approach exposes the jugular foramen anteriorly, and it may be suitable for selected tumors that extend along the petrous portion of the internal carotid artery, through the eustachian tube, or the cancellous portion of the petrous apex [3]. A preauricular skin incision is performed extending across the zygomatic process of the temporal bone into the cervical region. A fronto-temporal craniotomy with or without the superior and lateral orbital rim removal is performed. The middle cranial fossa floor is removed from lateral to medial until the carotid canal is reached. Both the eustachian tube and tensor tympani muscle are removed. Removal of the styloid process allows anterior mobilization of the internal carotid artery and access to the clival region. The drilling of Kawase’s triangle allows resection of the petrous apex and provides a corridor to the posterior fossa [8,25,27].

In 1978, Fisch described three types of tumor classification and infratemporal fossa approaches to the lateral skull base. Type A tumors are jugulotympanic paragangliomas. Type B tumors are jugular paragangliomas with no destruction of the bone, and type C tumors are jugular paragangliomas with the destruction of the infralabyrinthine compartment of the temporal bone. The Type A approach allows access to the temporal bone in its infralabyrinthine component and is suitable for jugular foramen tumors. The external auditory canal is transected at the bone-cartilage junction [28,29]. However, this approach is often combined with a lateral approach to access tumors with more anterior extension (Type B or C Fisch’s classification of paraganglioma tumors), requiring two-staged surgeries with a bigger chance of auditory loss and facial nerve damage [29].

### 3.3. Combined Approach

#### Combined Transmastoid Retro- and Infralabyrinthine Transjugular Transcondylar Transtubercular Transcervical Approach

From an anterolateral perspective, we can use a combined approach to reach total exposure of the jugular foramen in a single-stage surgery. This approach allows radical resection of tumors around the jugular foramen, the lower clivus, and the high cervical region. It combines the transmastoid, retro- and infra-labyrinthine transcondylar transtubercular and transcervical approaches. Multidirectional angles of attack and working corridors can be performed, including suprajugular, transjugular, and infrajugular exposures, maximizing the advantages of the abovementioned approaches. Both intracranial and extracranial tumors can be removed in a one-stage procedure. Blind sac closure of the external ear canal can be performed selectively as indicated (tumor extension into middle ear), and permanent facial nerve re-routing and mandibular translocation are generally unnecessary, minimizing postoperative complications. Nevertheless, these maneuvers can be performed selectively based on the indicated pathology. Furthermore, access to the lower clivus is facilitated by anterior translocation of the vertical portion of the internal carotid artery and inferior translocation of the lower cranial nerves, if needed [9,30].

This complex approach to entirely expose the jugular foramen can be simplified stepwise: 1. Postauricular C-shaped infratemporal incision; 2. Retrolabyrinthine mastoidectomy; 3. High cervical exposure; 4. Skeletonization and anterior translocation of the facial nerve; 5. Lateral suboccipital craniotomy and transcondylar transtubercular exposure; 6. Exposure of the internal jugular vein, jugular bulb, and sigmoid sinus; 7. Intradural exposure (for tumors with intracranial extension) [9,30] Figure 1B-3.

The patient is positioned supine with the head turned contralaterally. A retroauricular curvilinear C-shaped skin incision 2 to 3 cm posterior to the upper border of the ear is performed (Figure 1B). It continues posteroinferiorly into the neck crossing the anterior boundary of the sternocleidomastoid muscle and reaching underneath the mandibular angle. Prior to mastoidectomy, the entire body and tip of the mastoid, the spine of Henle, the posterior end of the root of the zygoma, the supramastoid crest, and the asterion must be exposed [9,30].

From that point, we skeletonize the semicircular canals, fallopian canal, sigmoid sinus, and jugular bulb (Figure 2A). The extracranial sections of the lower cranial nerves, the internal carotid artery, and the internal jugular vein are identified using a high cervical exposure (Figure 2B). After dividing the subcutaneous tissue and platysma muscle, the posterior angle of the mandible and the anterior boundary of the sternocleidomastoid muscle are identified by blunt dissection. Later, the facial nerve in the fallopian canal is totally skeletonized with a diamond burr from the genu to the stylomastoid foramen. The fallopian bridge technique involves leaving the facial nerve invested in its protective bone shell to avoid facial nerve damage. The mastoid tip is then removed with a high-speed drill to decompress the facial nerve from the stylomastoid foramen [9,30].

Exposure of the deeply seated suboccipital triangle provides a crucial anatomical landmark for this portion of the approach. It is important to open it by separating the superior and inferior oblique muscle insertions from the transverse process of C1 and reflecting them medially. The dorsal ramus of the C1 nerve root and the V3 horizontal portion of the vertebral artery can be found in this triangle. A lateral suboccipital craniectomy is then performed. Extradural reduction in the occipital condyle and jugular tubercle are the critical maneuvers of this step (Figure 3A,B). Removal of the posterior and medial one third of the occipital condyle is generally enough to increase the surgical corridor to the ventral foramen magnum [9,30].

The tumor mass is generally palpable within the venous structures after entire exposure of the sigmoid sinus, jugular bulb, and internal jugular vein. We can then coagulate all of the tumor’s arterial feeders and ligate the internal jugular vein slightly inferior to the tumor bulk. A suture ligature is used to occlude the sigmoid sinus immediately above the tumor. Alternatively, an endoluminal occlusion using gelfoam packing can also be used to avoid any durotomies (for entirely extradural tumors). This involves incising the lateral wall of the sigmoid sinus and inserting gelfoam pledgets proximally into the sigmoid sinus. Care is taken to avoid occluding the transverse sigmoid junction where the vein of Labbe enters. Control of back bleeding from the inferior petrosal sinus is controlled by injecting a flowable hemostatic matrix (Surgiflo, Ethicon, Inc., Bridgewater, NJ, USA) distally towards the jugular bulb. The lateral wall of the internal jugular vein is incised and the intraluminal tumor within the IJV, jugular bulb and sigmoid sinus is removed. If needed, for intradural pathology, the posterior fossa dura including the retrosigmoid, transsigmoid, and/or presigmoid dural incision, can be made to access the intradural portion of the tumor [9,30].

### 3.4. Representative Cases

The selection for the approach for jugular foramen tumors must be tailored case by case, considering the tumor’s configuration, the patient’s neurological status, and the neurosurgeon’s experience. Here, we present four cases in which the combined approach can be used to treat large complex jugular foramen tumors.
Case 1:

A 69-year-old female patient presented to our institution with progressive headaches, dysphagia, dysphonia, hearing loss, and severe gait ataxia from a left jugular foramen paraganglioma invading the jugular bulb and internal jugular vein with intradural compression of the brainstem (Figure 4A,B). After preoperative tumor embolization, the tumor was resected via a combined transmastoid infralabyrinthine transjugular transcervical approach. The internal jugular vein was ligated, and the sigmoid sinus was occluded endoluminally. The jugular bulb and vein were opened to remove the intraluminal invasion by the tumor. The intradural tumor was then removed to decompress the brainstem. Postoperatively, the patient remained at her neurological baseline with no new cranial nerve deficits. Severe gait ataxia improved. Postoperative MRI showed gross total resection of the tumor (Figure 4C,D).
Case 2:

A 24-year-old female patient presented with progressive headaches, dysphagia, and right-sided weakness from a left jugular foramen meningioma invading the jugular bulb and internal jugular vein with compression of the brainstem (Figure 5A,B). The tumor was resected via a combined transmastoid infralabyrinthine transjugular transcervical approach. In this case, blind sac closure was unnecessary since the tumor had no extension into the middle ear. The retrosigmoid corridor was opened to remove the intradural portion of the tumor. Then, the internal jugular vein was ligated, and the sigmoid sinus was occluded endoluminally (Figure 6A–D). The jugular bulb and vein were opened to remove the intraluminal invasion by the tumor. Postoperatively, the patient was neurologically intact with no cranial nerve deficits. Postoperative MRI showed gross total resection of the tumor (Figure 5C,D).
Case 3:

A 51-year-old male presented with progressive dysphagia and significant weight loss with 9th through 12th cranial nerve palsies. MRI demonstrated a giant craniocervical junction chordoma with jugular foramen invasion, extension into the right parapharyngeal space and intradural cervicomedullary junction, and near-complete erosion of the right occipital condyle (Figure 7A–C). The tumor was resected via a right combined infralabyrinthine transjugular transclival transcervical approach. The internal jugular vein was ligated, and the sigmoid sinus was occluded endoluminally with gelfoam. The jugular bulb was opened to reveal intrabulbar chordoma. A gross total tumor resection in all the invaded spaces was achieved, and the pre-clival dura was resected (Figure 7D,F). Multi-layered reconstruction with an alloderm graft and fat graft was performed. A second-stage occipital cervical fusion was performed because of the craniocervical instability caused by tumor invasion into the right occipital condyle. Postoperatively, the patient remained at his preoperative neurological baseline without any new neurological deficits.
Case 4:

A 34-year-old female presented with left-sided hearing loss, left vocal cord paralysis, and severe gait ataxia from a left dumbbell jugular foramen schwannoma involving the cerebellopontine angle (Figure 8A,B). A gross total resection was achieved via a combined infralabyrinthine, trans-sigmoid transjugular approach (Figure 8C,D). Postoperatively, the patient improved gait function and incurred no new cranial nerve deficits.

## 4. Discussion

The approach selection for jugular foramen tumors can vary significantly according to the lesion’s type, size, and configuration. Also, experience and extensive knowledge of the region’s anatomy are critical for achieving optimal clinical results. We aim to have a philosophy of maximal safe resection, avoiding neurovascular injury, and optimal preservation of functioning cranial nerves. A more aggressive approach can be considered in cases with pre-existing irreversible cranial nerve palsies [31].

The posterolateral approaches, especially the retrosigmoid approach, are indicated for tumors predominantly intradural with little or no extension to the extradural compartment. The anterolateral approaches can provide an excellent pathway when the tumor is extending to the infratemporal ICA or when the middle fossa approach is needed for tumors extending to the petrous apex.

The combined transmastoid retro- and infralabyrinthine transjugular transcondylar transtubercular transcervical approach described above with a high cervical C-shaped retroauricular incision allows a single-staged radical resection of large complex jugular foramen tumors. This approach has the advantage of delivering a total exposure of the jugular foramen through various attack angles. Without transection of the external ear canal, permanent facial nerve re-routing, or mandibular translocation, the infratemporal carotid artery can be exposed [9].

The permanent transposition of the facial nerve is a regular step during most of the trans-labyrinthine approaches [24]. After opening the stylomastoid foramen, the bone in the pre-facial area needs to be drilled. This drilling should be performed in the area that corresponds to the base of the styloid process. When this portion is drilled, it results in the detachment of the process. This detachment and the further permanent re-routing of the facial nerve provide exposure to the carotid canal, which contains the vertical C7 segment of the ICA [24,28,29]. Nevertheless, other authors have advocated performing an anterior, slightly vertical translocation of the facial nerve instead of permanent rerouting with less chance of facial nerve palsy when there is no need to expose the infratemporal segment (C7) of the carotid artery [9,31]. Translocating the facial nerve can often lead to temporary facial nerve palsy, so this technique, as well as permanent rerouting of the nerve, must be used with caution. It is important to keep the periosteum, which surrounds the facial nerve and contains the blood supply to the nerve, intact to ensure optimal preservation of the facial nerve [9].

Surgery of jugular foramen tumors often requires vascular control of the jugular venous system by ligating the internal jugular vein and sigmoid sinus to allow intrabulbar access. Occlusion of the sigmoid sinus traditionally involves pre- and retrosigmoid durotomies to allow the application of vascular clips or suture ligatures, with the risk of postoperative cerebrospinal fluid (CSF) leakage and pseudomeningoceles. An endoluminal sigmoid sinus occlusion with the gelfoam technique that is entirely extradural can avoid any durotomies that can result in postoperative CSF leaks. It can be performed by placing pieces of gel foam endoluminally into the proximal sigmoid sinus after the jugular vein is tied off. Care must be taken to avoid occlusion of the venous outflow of the vein of Labbe at the transverse-sigmoid junction to avoid temporal lobe venous infarction.

CSF leakage rate in jugular foramen tumors after surgery may vary according to the tumor size, type, configuration, intradural or extradural localization, and extension. There are several variables that must be taken into consideration when comparing one approach to another regarding CSF occurrence. There is little information in the literature because most of the case series presented have several approach selections, and none of them are designed to compare CSF outcomes from one approach to another. Nevertheless, some authors have reported low rates of CSF leaks, from 0% [10] in the retrosigmoid approach to higher rates of 4.5% [32,33], 5.3% [34], and 10% [35,36] in the translabyrinthine approach for other types of tumors. More studies should be conducted to give a more accurate CSF leak rate for each approach and each type of jugular foramen tumor.

Nevertheless, there are general pearls and tricks that the surgeon must follow in other to decrease the chances of CSF leakage. The first step involved is to perform a watertight dural repair using the dural sling technique with autologous fascia lata [37]. This involves suturing a piece of autologous fascial lata graft to the borders of the dural defect with interrupted 4-0 Nurolon (Ethicon) sutures producing a dural sling. Next, Tachosil (fibrin sealant on a collagen carrier) is placed on the edges, creating an additional layer of protection and further reinforcing the repair. After the repair has been reinforced, a fat graft is placed on top of the fascial sling and fills the mastoidectomy defect. Creating a facial sling prevents the fat from being in direct contact and compression with the intradural structures such as the brainstem and cranial nerves, and reduces the chances of fat necrosis, subarachnoid fat embolism and lipoid meningitis [38,39,40]. Then, a Medpor titanium plate may be used to reconstruct the bony defect and also buttresses the fat graft against the dural closure to further prevent a csf leak. It is important to seal off the entrance of the mastoid antrum into the middle ear with a bone wax plate and also to wax off any air cells as these may sometimes provide alternate accessory pathways to the middle ear. We usually reinforce this with another layer of calcium phosphate bone substitute (Hydroset, Stryker). The remaining dead space in the mastoid cavity is filled with fat graft and buttressed with a Medpor Titan plate. Additionally, lumbar drainage is performed for 3–5 days postoperatively at 5–10 cc per hour to help reduce the risk of pseudomeningocele formation and CSF leakage [37].

Endoscopic assistance can also be performed to assess the degree of tumor resection and improve visualization of the lower cranial nerves after removing the tumor. The need to perform a tracheostomy and/or gastrostomy is tailored on a case by case basis and will depend mostly on postoperative assessment with video laryngoscopy and modified barium swallow. Patients with dysphonia, either pre-existing preoperatively or newly developed postoperatively, may require vocal cord injections to improve voice function. Additional maneuvers with botox injection or esophageal balloon dilatation can be considered based on the assessment by our laryngology team.

A multidisciplinary team comprising neurosurgeons, otolaryngologists (including neuro-otologist, laryngologist, head and neck surgeon), radiation oncologist, and neurinterventionalist is critical for management of complex jugular foramen tumors. We encourage all neurosurgeons to first have extensive anatomical knowledge of the jugular foramen through cadaveric dissections or 3D simulations before attempting to perform these types of approaches.

## 5. Conclusions

Surgery for jugular foramen tumors can be complex and laborious. Each case should be thoroughly evaluated to determine the best strategy for the patient, allowing the surgeon to achieve maximal safe removal of the tumor while avoiding neurological complications. Our workhorse is the combined transmastoid retro- and infralabyrinthine transjugular transcondylar transtubercular high cervical approach which can be used to resect difficult jugular foramen tumors such as paragangliomas, schwannomas, meningiomas, and chordomas. Total exposure of the jugular foramen can be achieved, and multidirectional approaches can be performed, including suprajugular, infrajugular, and transjugular corridors. A multidisciplinary team and extensive understanding of the surgical anatomy are essential to offer the patient with the greatest chance of optimal postoperative outcomes.

## Figures and Tables

**Figure 1 brainsci-14-00182-f001:**
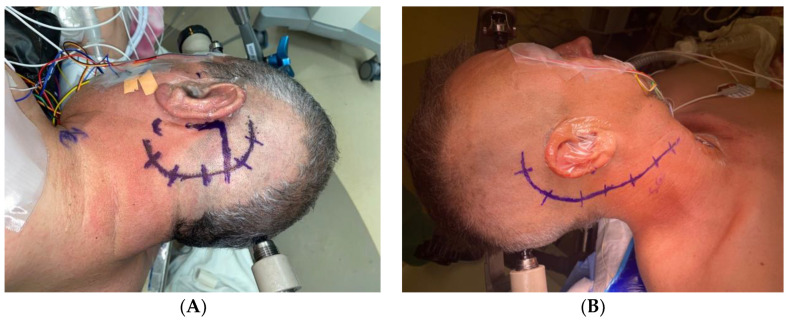
(**A**). Retrosigmoid C-shaped retroauricular skin incision posterior and parallel to the outline of the pinna (**B**). Combined approach right-sided C-shaped retro-auricular incision. The incision is started approximately 2 to 3 cm posterior to the upper border of the ear. It continues posteroinferiorly into the neck over the anterior border of the sternocleidomastoid muscle and under the mandibular angle.

**Figure 2 brainsci-14-00182-f002:**
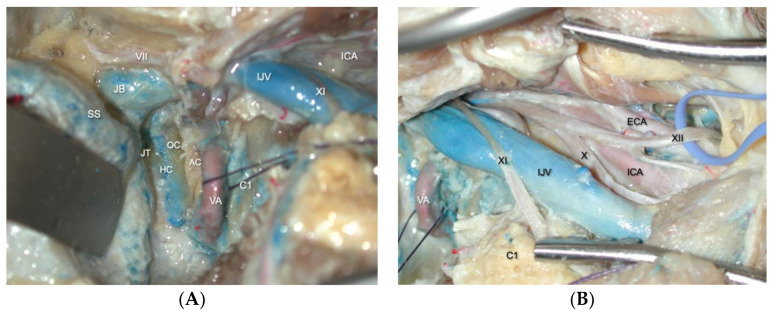
(**A**) Lateral view of the combined transmastoid retro- and infralabyrinthine transjugular transcondylar transtubercular high cervical approach, reflecting the posterior fossa dura. The JB passing through the jugular foramen. The IJV descends along the ICA with the lower cranial nerves. The vertebral artery ascends through the transverse process of C1 and usually passes behind the atlantal condyle. (**B**) Extracranial transcervical perspective of the glossopharyngeal, vagus, accessory, and hypoglossal nerves. The IX and XII nerve pass anteriorly along the lateral surface of the ICA. The XI nerve descends posteriorly across the lateral surface of the IJV. The vagus descends inferiorly within the carotid sheath. *AC., atlantal Condyle. C1., atlas. ECA., external carotid artery. HC., hypoglossal canal. ICA., internal carotid artery. IJV., internal jugular vein. JB., jugular bulb. JT., jugular tubercule OC., occipital condyle. PFD., posterior fossa dura. SS., sigmoid sinus. VA., vertebral artery. VII., facial nerve. XI., accessory nerve. IX., glossopharyngeal nerve. X., vagus nerve. XII., hypoglossal nerve*.

**Figure 3 brainsci-14-00182-f003:**
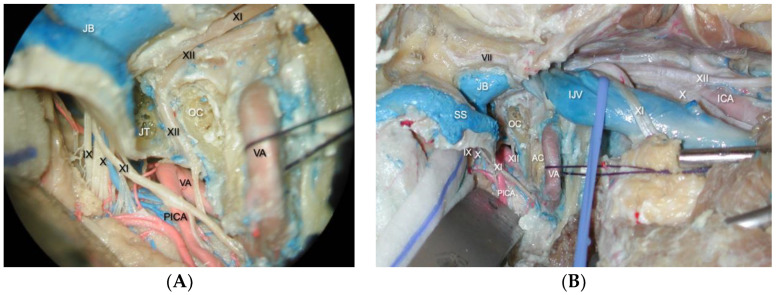
(**A**). Intradural and extradural views of the lower cranial nerves passing through the jugular foramen. The glossopharyngeal, vagus, and accessory nerves arise from the medulla in the postolivary sulcus and pierce the dural roof of the jugular foramen to pass through it. The IX nerve enters the jugular foramen through the glossopharyngeal meatus, and the X and XI nerves through the vagus meatus. The PICA arises from the posterior or lateral surfaces of the VA. The XII exits through the hypoglossal canal above the OC. (**B**) Final view of the combined approach with high cervical exposure. From the intradural perspective, the lower cranial nerves leaving the medulla and enter the jugular foramen. Hypoglossal nerve exits through the hypoglossal canal. Vertebral artery below the OC. Sigmoid sinus empties into the jugular foramen after coursing down the sigmoid sulcus, crossing the occipitomastoid suture at the site of the jugular bulb. From the jugular bulb, the flow is directed downward into the IJV. *AC., atlantal condyle ICA., internal carotid artery., IJV., internal jugular vein. JB., jugular bulb. JT., jugular tubercule. OC., occipital condyle., PICA., postero-inferior cerebellar artery. SS., sigmoid sinus. VA., vertebral artery. VII., facial nerve. IX., glossopharyngeal nerve. X., vagus nerve. XI., accessory nerve. XII., hypoglossal nerve*.

**Figure 4 brainsci-14-00182-f004:**
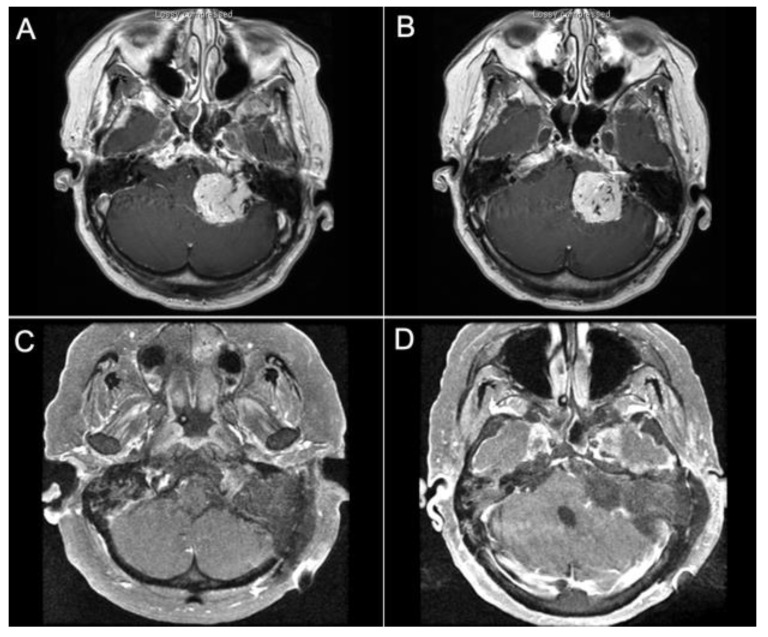
Pre-op and Post-op MRI demonstrating a left jugular paraganglioma (glomus jugulare). (**A**,**B**) The images show a large jugular paraganglioma that invaded into the cervical IJV and had significant extension intradurally into the cerebellopontine angle with compression of the brainstem. (**C**,**D**) The images show gross total resection of the tumor by the combined transmastoid retro- and infralabyrinthine transjugular transcondylar transtubercular high cervical approach.

**Figure 5 brainsci-14-00182-f005:**
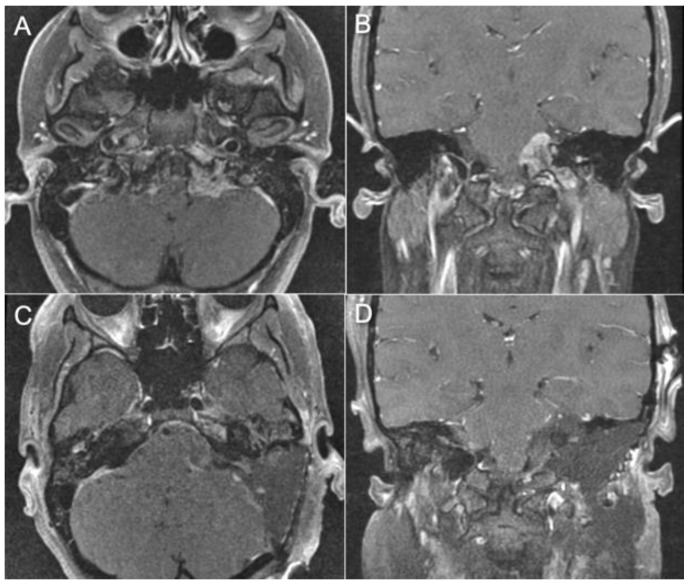
Pre-operative and post-operative MRI views of a left jugular foramen meningioma invading the internal jugular vein. (**A**,**B**) The images show T1 gadolinium- enhanced images of a homogenous mass in the jugular foramen with intradural extension. (**C**,**D**) The images show gross total resection of the tumor by the combined transmastoid retro- and infralabyrinthine transjugular transcondylar trans tubercular high cervical approach.

**Figure 6 brainsci-14-00182-f006:**
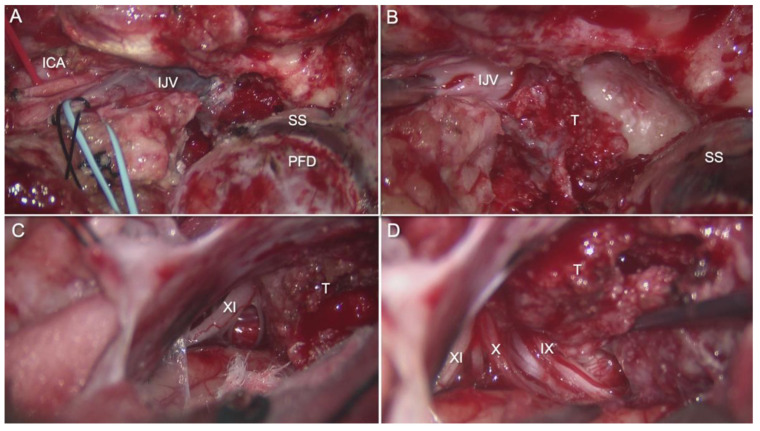
(**A**) Left-sided exposure of jugular foramen via extended anterolateral infralabyrinthine transjugular approach for resection of jugular foramen meningioma. (**B**) After tying off the IJV and endoluminal occlusion of the sigmoid sinus, the lateral wall of the sigmoid sinus, jugular bulb and internal jugular vein are excised to expose the intraluminal tumor in the jugular bulb. (**C**,**D**) Retrosigmoid exposure of the intradural portion of the tumor at the jugular fossa. The tumor is carefully dissected from the lower cranial nerves. ICA., internal carotid artery. IJV., internal jugular vein. PFD., posterior fossa dura. SS., sigmoid sinus. T., tumor.

**Figure 7 brainsci-14-00182-f007:**
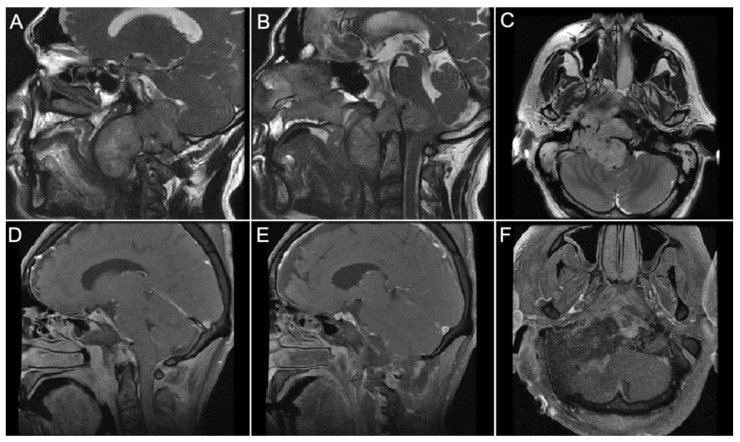
Pre-op and Post-op MRI views of an extensive cranio-cervical chordoma invading the jugular foramen, cerebello-medullary cistern and parapharyngeal space. (**A**–**C**) The images show a T2 MRI signal with a large lobulated mass centered on the right parapharyngeal space with intradural and extradural extension. The tumor is compressing the anterior pons and the airway deviated to the left side. (**D**–**F**) The images show a gross total resection of the tumor using the combined transmastoid retro- and infralabyrinthine transjugular transcondylar transtubercular high cervical approach.

**Figure 8 brainsci-14-00182-f008:**
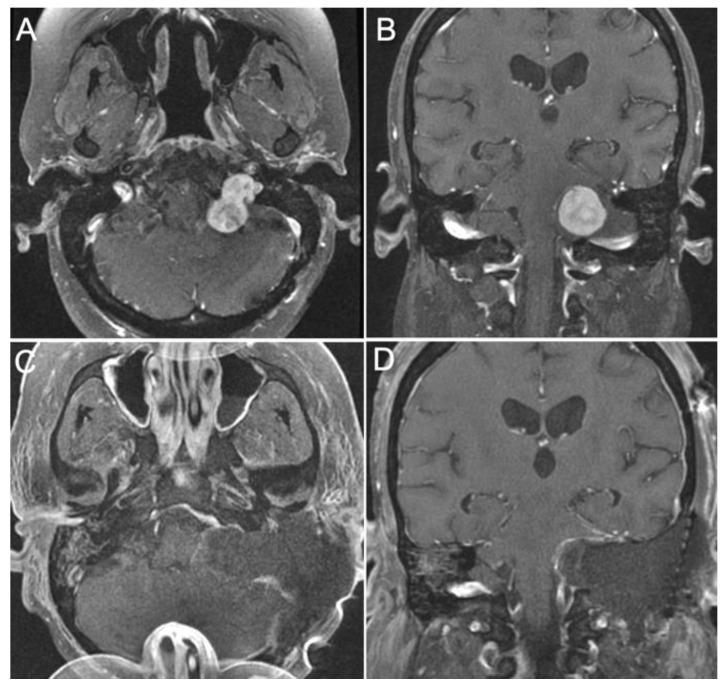
Pre-op and post-op MRI views of a jugular foramen schwannoma. (**A**,**B**) The images show a gadolinium enhanced T1 weighted MR images with a well- circumscribed dumbbell shaped Schwannoma invading the jugular foramen. (**C**,**D**) The images show gross total resection of the tumor using a combined transmastoid retro- and infralabyrinthine transjugular transcondylar transtubercular high cervical approach.

**Table 1 brainsci-14-00182-t001:** Surgical approaches classification for jugular foramen tumors.

Anterolateral	Posterolateral	Combined
Postauricular transtemporal approach	Retrosigmoid approach	Combined transmastoid retro- and infralabyrinthine transjugular transcondylar trans tubercular high cervical approach
Preauricular Subtemporal infratemporal approach	Far-Lateral approach -Transcondylar approach -Supracondylar approach -Paracondylar approach	

## Data Availability

No new data were created or analyzed in this study. Data sharing is not applicable to this article.

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
