# Peer review of "Jugular Foramen Tumors: Surgical Strategies and Representative Cases"

_brainsci, 2024, doi:10.3390/brainsci14020182_

Round 1

Reviewer 1 Report

Comments and Suggestions for Authors

In my opinion, there are several critical points that readers would be interested in knowing that are not addressed in the article:

1. Know more precisely, in the Far Lateral Approach, when it is necessary to drill the occipital condyle.

2. In this same approach, it would be very useful to know when it is necessary to fix the craniovertebral junction after surgery.

3. Is it necessary to transpose the facial nerve in transtemporal approaches? When yes? When do not?

4. What is the risk of cerebrospinal fluid leak in each approach and what measures can be followed to avoid it?

5. When is it recommended to perform a tracheostomy and/or grastrostomy in these cases? When to do it preventively before surgery and when it is better to wait to evaluate the post-operative clinical result?

The present work would be an excellent publication for a textbook, but not for a scientific journal, since it does NOT provide any new information to that already known in multiple similar publications. Considering the great experience of the group of authors and co-authors on the topic, two options are suggested:

1.    Present your experience with a sufficient number of cases for each approach. They could be divided into four different series: patients with lesions of the jugular foramen operated on using A) Far Lateral approach; B) Transtemporal approach; C) Pre-auricular approach and D) Combined approaches.

2.    Present your experience regarding the histological variety of jugular foramen tumors divided into three series: A) Paragangliomas; B) Schwannomas; C) Meningiomas and D) Chordomas and chondrosarcomas.

In either of these two options, the authors could publish 3 to 4 different articles, with more refined and valuable information, which would surely be a true contribution to international literature.

Author Response

Thank you for your valuable feedback and comments that will help us upgrade our article. The intent and purpose of this submission was an editor’s request for a review article of surgical approaches to the jugular foramen. Therefore, our manuscript was presented in the current fashion.

  • : The far lateral approach is a surgical technique that involves extending further laterally from the lateral suboccipital approach. 1 2 3 It also can include a transcondylar, supracondylar, or paracondylar extension with a further increase in the working space along the anterior border of the foramen magnum, jugular tubercle area, and posterior margin of the jugular foramen, respectively. 4 5 As stated in the article, the degree of drilling the occipital condyle will depend on the access area needed. The para-condylar extension involves partial drilling of the lateral portion of the occipital condyle and the mastoid tip, and it is the approach choice for dumbbell schwannomas and paragangliomas of the jugular foramen. 1 2 4 The transcondylar is performed by drilling the posterior condyle to allow space to the anterior brainstem.3  The percentage of condyle to be removed can vary from only the posterior one-third with no instability of the craniovertebral junction to as far as the posterior half, which will require fixation of the craniovertebral junction due to instability of the atlanto-occipital joint. 6 The complete trans-condylar approach is done with 2/3 of the condyle being removed and will give complete access to the medullary area. 1

The preservation of the occipital condyle and the C-1 lateral mass, as well as the attachments of the alar and transverse ligaments to the anterior one-third of the occipital condyle and the anterior one-third of the C-1 lateral mass, are among the key variables that are responsible for maintaining the stability of the occipitocervical junction.4 2 An occipital-cervical fusion operation is required when the integrity of these structures has been impaired, whether through a complete trans-condylar approach, a transplacental approach, or tumor destruction of these areas. Fusion is usually performed as a second-stage operation and when there is no sign of cerebrospinal fluid leakage. This will be added to our article.

  1. The permanent transposition of the facial nerve is a regular step during most of the trans-labyrinthine approaches. After opening the stylomastoid foramen, the bone in the pre-facial area needs to be drilled. This drilling should be done in the area that corresponds to the base of the styloid process. When this portion is drilled, it results in the detachment of the process. This detachment and the further permanent re-routing of the facial nerve provide exposure to the carotid canal, which contains the vertical C7 segment of the ICA. 7 8 9. Nevertheless, other authors have advocated performing an anterior, slightly vertical translocation of the facial nerve instead of permanent rerouting with less chance of facial nerve palsy when there is no need to expose the infratemporal segment (C7) of the carotid artery. 10 11 Translocating the facial nerve can often lead to temporary facial nerve palsy, so this technique, as well as permanent rerouting of the nerve, must be used with caution. It is important to keep the periosteum, which surrounds the facial nerve and contains the blood supply to the nerve, intact to ensure optimal preservation of the facial nerve. 10 We will clarify this in our article.

  1. CSF leakage in jugular foramen tumors after surgery may vary according to the tumor size, type, configuration, intradural or extradural localization, and extension. There are several variables that must be taken into consideration when comparing one approach to another regarding CSF occurrence. There is little information in the literature because most of the case series presented have several approach selections, and none of them are designed to compare CSF outcomes from one approach to another. Nevertheless, some authors have reported low rates of CSF leaks, from 0% in the retro-sigmoid 12 approach to 4.5%13 and 5.3%14in the trans-labyrinthine combined approach. More studies should be done in order to give a more accurate CSF leak for each approach and each type of jugular foramen tumor. This will also be clarified in our article.

  1. In our clinical experience, the need to perform a tracheostomy and/or gastrostomy must be tailored case by case and will depend mostly on post-operative results, how the patient responds to the manipulation of the lower cranial nerves, how well preserved the lower cranial nerves were during surgery, and how much involvement of the cranial nerves existed before surgery. Preventing tracheostomy and or gastrostomy is not usually performed. Using neurostimulation during surgery can help improve post-operative lower cranial nerve results. The post-operative results are better evaluated…

We thank the reviewer for the suggestion of providing our clinical experience since we know it will be a great contribution to international literature. Nevertheless, the present work is a comprehensive review of the jugular foramen tumors and their approaches with illustrative cases to facilitate surgeons how to guide their selection approach based on its advantages and disadvantages; it doesn’t have the aim of presenting clinical results.

References:

  1. Sabino Luzzi, Alice Goitta Lucifero, Nunzio Bruno, Matias Baldoncini, Alvaro Campero, Renato Galzio. Far Lateral Approach. Acta Biomed. 2021;92(4):e2021352. doi:10.23750/abm.v92iS4.12823
  2. Chaddad-Neto F, Doria-Netto HL, De campos-Filho JM, Reghin-Neto M, Rothon Jr AL, De Oliveria E. The Far-lateral craniotomy: Tips ans Tricks. Arq Neuropsiquiatr. 2014;72(9):699-705. doi:10.1590/0004-282x20140130.
  3. Akihito Sato, Sakyo Hirai, Yoshiki Obata, Taketoshi Maehara, Masaru Aoyagi. Muscular-Stage Dissection during Far Lateral Approach and Its Transcondylar Extension. J Neurol Surg B Skull Base. 2018;79(Suppl 4):S356-S361. doi:10.1055/s-0038-1668518
  4. Salas E, Sekhar LN, Ziyal IM, Caputy AJ, Wright DC. Variations of the extreme-lateral craniocervical approach: anatomical study and clinical analysis of 69 patients. J Neurosurg (Spine 2). 1999;90:206-219.
  5. Albert L. Rhoton Jr. The Far-Lateral Approach and Its Transcondylar, Supracondular, and Paracondylar Extensions. Neurosurgery. 2000;47(3). doi:10.1097/00006123-200009001-00020
  6. Babu RP, Sekhar LN, Wright DC. Extreme lateral transcondylar approach: technical improvements and lessons learned. J Neurosurg. 1994;81:49-59.
  7. Takanori Fukushima, Yoichi Nonaka. Fukushima Manual of Skull Base Dissection. Third Edition. AF-Neuro Video, Raleigh, NC; 2010.
  8. Ugo Fisch. Infratemporal fossa approach to tumours of the temporal bone and base of the skull. The Journal of laryngology and otology. 1978;92(11):949-967. doi:10.1017/s0022215100086382
  9. Ugo Fisch, Harold C. Pillsbury. Infratemporal Fossa Approach to Lesions in the Temporal Bone and Base of the Skull. Archives of otolaryngology. 1979;105(2):99-107. doi:10.1001/archotol.1979.00790140045008
  10. James K. Liu, Tetsuro Sameshima, Oren N. Gottfried, William T. Couldwell, Takanori Fukushima. The combined Transmastoid Retro- and Infralabyrinthine Transjugular Transcaondylar Transtubercular High Cervical approach for resection of Glomus jugulare Tumors. Neurosurgery. 2006;59(1 Suppl 1):ONS115-25. doi:10.1227/01.NEU.0000220025.81500.8D
  11. Ossama Al-Mefty, Aramis Teixeira. Complex tumors of the glomus jugulare: criteria, treatment, and outcome. Journal of neurosurgery. 2002;97(6):1356-1366. doi:10.3171/jns.2002.97.6.1356
  12. Matsushima K, Kohno Michihiro, Nakajima N, et al. Retrosigmoid Intradural Suprajugular Approach to Jugular Foramen Tumors with Intraforaminal Extension: Surgical Series of 19 Cases. World Neurosurgery. 2019;125:e984-991. doi:10.1016/j.wneu.2019.01.223.
  13. Sanna M, Bacciu A, Falcioni M, Taibah A. Surgical management of jugular foramen schwannomas with hearing and facial nerve function preservation: a series of 23 cases and review of the literature. Laryngoscope. 2006;116(12):2191-2204. doi:10.1097/01.mlg.0000246193.84319.e5
  14. Orphée Makiese, Chibbaro Salvatores, M. Marsella, P. Tran Ba Huy, B. George. Jugular Foramen Paragangliomas: Manegement, outcome and avoidance of complications in a series of 75 cases. Neurosurg Rev. 2012;35:185-194. doi:10.1007/s10143-011-0346-1

Reviewer 2 Report

Comments and Suggestions for Authors

Dear Authors,

I found your study, regarding the surgical approaches for jugular foramen tumors, interesting and useful. Although rare, I agree with you that they pose a significant challenge to any surgical team due to complex local anatomy and of any approach might be chosen.

I have two ovservations:

A more extensive iconography would be helpful, and more consistent. Only one approach has an image with the positioning and the incision, and only two details of the anatomy of the region. 

As it is, the paper suggests that if we plan very carefully and have a thorough knowledge of the anatomy these tumors can be safely removed. While I completely agree, I believe that a discussion of the complications and their management is mandatory.

Comments on the Quality of English Language

The English language usage seems mostly correct. Minor problems can be easily solved on an additional reading. 

Author Response

Thank you for your suggestions and feedback.

  1. We will add a more extensive iconography for each approach since it is a comprehensive review of the literature, and it will help enrich our article and its aim.
  2. We totally agree that a better discussion of the complications and their management should be conducted, including CSF leakage rate, how to decrease lower cranial nerve injury, and how to maximize a safe resection.

Reviewer 3 Report

Comments and Suggestions for Authors

Liu et al. summarize different surgical techniques to access and treat jugular foramen tumors. While the presentation of the approaches is very comprehensive, it seems repetitive, and the procedures described have been published extensively (by the senior author himself (Operative Neurosurgery, and more publications from Fukushima et al. ). The method section is very short and ultimately states that the manuscript is based on "the senior author's experience." If the surgical technique has been described, and the main focus is to describe its feasibility, why not include the treated patient cohort? Including a larger case series would increase its scientific soundness and provide information on the clinical outcome of such extensive approaches, improving the scientific value of the manuscript.

Furthermore, the data availability statement should be corrected (it states the instructions and hasn't been updated).

Author Response

We thank the reviewer for suggesting that we share our clinical experience since we know it will be an invaluable addition to the international literature. However, the intent and purpose of this submission was an editor’s request for a review article of surgical approaches to the jugular foramen. Therefore, our manuscript was presented in the current fashion, to provide a comprehensive review of jugular foramen tumors and their different approaches, with illustrative cases, to assist surgeons in advising the appropriate approach based on its advantages and disadvantages; and did not include granular clinical data. 

Round 2

Reviewer 1 Report

Comments and Suggestions for Authors

With all changes made to the paper, I think it can be published in this new form

Reviewer 3 Report

Comments and Suggestions for Authors

All the suggestions have been considered. The manuscript ist now suitable for publication.